# JNK Pathway in CNS Pathologies

**DOI:** 10.3390/ijms22083883

**Published:** 2021-04-09

**Authors:** Teresa de los Reyes Corrales, María Losada-Pérez, Sergio Casas-Tintó

**Affiliations:** Instituto Cajal (CSIC), Av. Doctor Arce 37, 28002 Madrid, Spain; tedlreyes@cajal.csic.es (T.d.l.R.C.); m.losada@cajal.csic.es (M.L.-P.)

**Keywords:** MAP kinase, cell signalling, glioblastoma, neurodegenerative disorders, CNS injuries

## Abstract

The c-Jun N-terminal kinase (JNK) signalling pathway is a conserved response to a wide range of internal and external cellular stress signals. Beside the stress response, the JNK pathway is involved in a series of vital regulatory mechanisms during development and adulthood that are critical to maintain tissue homeostasis. These mechanisms include the regulation of apoptosis, growth, proliferation, differentiation, migration and invasion. The JNK pathway has a diverse functionality and cell-tissue specificity, and has emerged as a key player in regeneration, tumorigenesis and other pathologies. The JNK pathway is highly active in the central nervous system (CNS), and plays a central role when cells need to cope with pathophysiological insults during development and adulthood. Here, we review the implications of the JNK pathway in pathologies of the CNS. More specifically, we discuss some newly identified examples and mechanisms of JNK-driven tumor progression in glioblastoma, regeneration/repair after an injury, neurodegeneration and neuronal cell death. All these new discoveries support the central role of JNK in CNS pathologies and reinforce the idea of JNK as potential target to reduce their detrimental effects.

## 1. Introduction

The CNS is exposed to stress stimuli during development and adulthood, and under pathological aggressions. These events trigger the Jun N-terminal kinase signalling pathway (JNK) as a mechanism to coordinate cellular responses to stress, and maintain tissue homeostasis. The JNK pathway includes a conserved mitogen-activated protein kinase (MAPK), which belongs to the stress-activated protein kinase (SAPK) group, a group of kinases that can be activated by any internal or external stimuli that cause cell stress. Furthermore, the JNK pathway in the CNS and other tissues can be activated by UV irradiation, glucose deprivation, DNA damage, heat stress, bacterial and viral infection, oxidative stress, inflammatory cytokines and growth factors [1,2,3,4,5,6,7,8,9,10].

The molecular MAPK cascade presents high homology from *Drosophila* to mammals (Figure 1) [11]. In *Drosophila,* JNK pathway-signalling is initiated by the interaction of the ligand protein Eiger (Egr), the unique TNF superfamily member of ligands [12,13], with TNF receptors (TNFRs) Grindewal (Grnd) [14], or Wengen (Wng) [15]. Ligand–receptor interaction initiates a cascade of phosphorylations that mediates the JNK signalling pathway [11]. In *Drosophila*, JNK is encoded by a single gene: *basket* (*Bsk*); this simplifies the genetic studies that have contributed to decipher the role of JNK under physiological and pathological stressful scenarios [16,17]. In mammals, the JNK cascade involves four kinases, and mitogens or cytokines induce MAP3K family activation [11]. MAPK cascade triggers cytosolic JNK dual phosphorylation and initiates the phosphorylation of cytoplasmic and nuclear proteins [18], including cytoskeletal and mitochondrial proteins, nuclear transcription factors, membrane proteins or nuclear hormone receptors [19]. Thus, gene expression derived from JNK activation leads to a variety of responses, depending on the cell type and the scenario [11]. In mammals, the JNK pathway is encoded by *jnk1*, *jnk2* and *jnk3* [20]. JNK1 and JNK2 are found ubiquitously, whereas JNK3 is restricted to the brain, cardiac smooth muscle and testis [20,21]. In particular, the JNK pathway is highly active in the CNS as compared with other tissues. Therefore, it emerges as a critical regulator of CNS cells under physiological and pathological conditions [22,23]. It has been demonstrated in mouse that different JNK isoforms undergo compensatory mechanisms during early brain development. Although a single deficiency of each JNK isoform is viable, the mutants still present different phenotypes, highlighting the importance of each individual enzyme (reviewed in [23]). Besides, each isoform has different temporal and regional expression patterns (reviewed in [21]).

The JNK signalling pathway mediates embryonic development; metabolism and growth; lifespan; programmed cell death; cell migration, repair and proliferation; immunity; and axonal transport [24,25,26,27,28]. In addition, the literature reveals a dual role of the JNK pathway in cell death and cell survival, depending on the cell type and the context [11]. This two-faded role is particularly important in CNS pathologies such as neurodegeneration and tumorigenesis, in which the cellular stress-associated signals are increased [29,30]. Likewise, JNK signalling is involved in neurogenesis, axonal growth, axonal transport, brain metabolism, animal behavior, neurulation, migration and axon–dendritic architecture in different species [21,22,23,31,32].

Here, we describe how neurons and glia regulate stressful environments such as brain tumors, acute injuries or neurodegenerative diseases through the JNK pathway.

## 2. JNK in Glioblastoma Tumor Progression

Glioblastoma multiforme (GB) is the most frequent and aggressive malignant primary brain tumor of the CNS in humans [33,34]. Given the aggressiveness of the tumor and its resistance to current treatments, patients have a poor prognosis, and the median survival is limited to 12–15 months after the diagnosis [33,34,35]. Among the mechanisms involved in GB malignancy, JNK pathway activation is a hallmark associated with cancer stem-cell-like properties, cancer-initiating potential and glial proliferation [30,36,37].

The most common genetic lesions in GB patients include the epidermal growth factor receptor (EGFR) mutations, loss of PTEN (antagonist of PI3K) and the mutation of the catalytic activity domain of PI3K [35,38,39]. Li et al. collected samples of different glioma grades from patients exposed to surgery to study the JNK activity’s correlation within the tumor progression [40]. Although in vitro and in vivo experiments suggested a positive correlation of EGFR and JNK activation with glioma grades [36,40], Li et al. revealed that 80% of EGFR-negative glioblastoma samples showed JNK activation [40]. This suggests that JNK pathway activation is sensitive to various inputs in GB cells. In addition, JNK signalling pathway modulated bypass resistance to receptor tyrosine kinase (RTK)-targeted therapies in GB [41].

Recent findings in *Drosophila* demonstrate that GB aggressiveness correlates with JNK pathway progressive activation [42]. First, the tumor induces neurodegeneration, followed by an infiltration through the brain with ultralong extended glial membrane protrusions known as cytonemes (mimicking mammal tumor microtubes, TMs), and GB cell proliferation [42,43,44]. The glial network mediates cell-to-cell communication, and promotes the exchange of molecules between neurons and GB cells. Consequently, the JNK pathway is activated in GB cells and promotes GB progression and infiltration [30,42,45].

JNK activation leads to an upregulation of JNK target genes, including matrix metalloproteases (MMPs) [30]. MMPs mediate extracellular matrix (ECM) degradation, and therefore facilitate TM infiltration [30]. Thus, the expansion of the TM network promotes communication among cells, maintaining a positive loop [30,42]. The *Drosophila* JNK ligand Egr is expressed in neurons under physiological conditions, but in GB samples, there is a progressive increase of JNK-pathway activation via the Grnd receptor by a relocalization of the ligand from neurons to glial cells [42]. Consistently with the tumor chronogram, the relocalization of Egr to glial cells coincides with the synapse loss, as a previous event to tumor infiltration and proliferation. More recently, it has been shown that GB cells produce Impl2, an insulin-pathway antagonist, as a mechanism to induce neuronal alterations including mitochondrial damage, contributing to neurodegeneration and tumor progression [45]. This strategy is also mediated by JNK activity that regulates *Impl2* levels of expression in GB cells [45]. Thus, GB cells show different strategies that converge into JNK signalling, and in turn, JNK mediates further GB expansion at the expense of neighbouring neurons (Figure 2).

The TM network is responsible for tumor aggressiveness, but also for conventional therapy resistance, thus it becomes a potential target to approach tumor malignancy [44]. Temozolomide (TMZ) is the most effective antiglioma agent so far; however the administration dose is restricted by the limited patient survival, and patients develop treatment resistance [46,47]. In vitro cell-culture experiments revealed that the JNK pathway mediates TMZ resistance to cytotoxicity [48]. Thus, JNK inhibitors emerge as potential candidates for GB pathogenesis, and have been proposed as a combined therapeutic opportunity with TMZ administration [37,48].

JNK-pathway inhibition through *Egr* or *Grnd* knockdown experiments rescue tumor proliferation and invasiveness in *Drosophila* [42]. Similarly, in vitro and in vivo experiments showed that the JNK inhibitors SP600125 and AS602801 impaired GB self-renewal and tumor-initiating potential [37,49]. SP600125 is a reversible JNK inhibitor reported as an anticancer clinical potential target due to its systemic administration and long-lasting and short-time selective effects as a JNK inhibitor [37]. However, JNK inhibitors’ doses and benefits/risk profiles limit human clinical trials. Moreover, it has been extensively described that JNK activity is cell- and tissue-specific based on isoform dependency and tumor stage [50]. Thus, the relevance of JNK isoforms in GB is not well understood; therefore, there are no approved treatments based on JNK inhibitors for GB patients. Although these molecules have emerged as new potential treatments in combination with conventional therapies, further research will shed light to understand the JNK pathway as an anticancer target.

## 3. JNK in Neurodegenerative Disorders

There are multiple stress stimuli that drive neurodegeneration, inducing programmed cell death, necrosis and autophagy (reviewed in [51]). Aging, accumulation of misfolded proteins, oxidative stress, inflammation and hypoxia are the classical stimuli that alter cell homeostasis and shift the balance into a stressful condition, in which cells tend to degenerate (reviewed in [51,52]). JNK3, the major isoform expressed in the CNS, contributes to neuronal cell degeneration in this context (reviewed in [29,53]). JNK3 also plays a role in synapse dysfunction, one of the first events in neurodegenerative and neurodevelopmental brain diseases [54]. Therefore, JNK3 has been proposed to be involved in neuronal-cell-death mechanisms in neurodegenerative disorders, adding a new promising target to approach CNS illnesses.

### 3.1. Alzheimer’s Disease (AD)

AD is the most frequent neurodegenerative disorder, and is based on the aggregation of neurofibrillary tangles and Aβ peptide deposition in cortical and hippocampal neurons (reviewed in [55]). Aβ–JNK mechanisms attenuate the expression of antiapoptotic *Bcl-2* genes, required for the apoptosis of cortical neurons that occur in AD progression [56]. In addition, phosphorylated tau promotes tangle formation, and correlates with JNK enzyme activity [57]. It has been shown that before early onset of the pathology, the JNK signalling pathway is activated in the excitatory postsynaptic neuronal dendritic spines, inducing synapse loss [54]. JNK3 phosphorylated levels correlate with the progression of AD [58,59]. In vitro and in vivo models of AD have demonstrated that JNK knockout reduces synapse loss, Aβ depositions and neuronal cell death [54,55,60]. All together, these results indicate that JNK signalling plays a central role in AD neurodegeneration, but JNK also has emerged as a potential biomarker for early diagnosis, and a potential target to prevent neuronal cell death [55]. The first synthesized therapeutic candidate was the SP600125 JNK inhibitor, which prevents neuronal cell death in vitro and in vivo in AD models [55]. However, several studies reported that this ATP-competitive inhibitor is not selective for JNK isoforms, and binds to other JNK nonrelated kinases [61]. Thus, the lack of specificity and the variety of toxicity degrees that it provokes require further research. Based on that, multiple new JNK inhibitors have been designed within recent years to target AD models [55,62]. However, specificity and toxicity of JNK inhibitors require further studies. JNK3 is the major isoform involved in AD pathology; therefore, it is plausible to hypothesize that the specific inhibition JNK3 isoform will limit these off-target effects in the future.

### 3.2. Parkinson Disease (PD)

JNK-pathway signals also mediate neuronal cell death in PD. In vitro and in vivo PD models revealed the induction of JNK3 activation in dopaminergic neurons of the substantia nigra after the administration of neurotoxins [63]. The involvement of JNK in the activation of apoptosis through the increase of the Bax/Bcl-2 ratio is well described in PD [64]. JNK suppresses the inhibiting functions of Bcl-2, indicating the ability to induce not only apoptosis but also autophagy [64]. Growing evidence suggests that the inhibition of the JNK pathway is a protective strategy against the neurodegeneration caused in PD [65,66]. Recent discoveries indicate that the receptor-interacting protein kinase 1 (RIPK1) is upregulated in PD in vitro and in vivo models. RIPK1 promotes cell apoptosis, necrosis, inflammation response, reactive oxygen species (ROS) production and mitochondrial dysfunction through the activation of the JNK pathway. Additionally, it has been recently shown that the RIPK1 inhibitor Nec-1s has neuroprotective effects against PD, as it inhibits the ASK1/JNK signaling pathway [67]. Finally, pharmacological activation of dopamine D1 receptors induces JNK phosphorylation in the striatal projection neurons, and JNK mediates dopamine transmission upon dopamine depletion, which has also been proposed as a potential therapy for PD [68]. Similarly, *Drosophila* PD models show JNK/Bsk phosphorylation [17]. Finally, the expression of a dominant negative form of JNK/Bsk in DA neurons rescues survival, neurodegeneration and locomotion impairment in PD [17,63].

### 3.3. Huntington’s Disease (HD)

HD is a neurodegenerative disorder characterized by the degeneration of projection neurons in the striatum and the cerebral cortex [69]. At the molecular level, HD is caused by expansion of a glutamine repeat (polyQ) in the N-terminus of the huntingtin protein (HTT) that provokes on protein aggregation and neuronal death [69]. Expression of polyQ-containing proteins has been previously shown to induce various cellular stress responses, including JNK. Furthermore, different studies have revealed the implication of aberrant HTT in the impairment of fast axonal transport (FAT), a neuronal challenge in neurodegenerative process [70]. Morfini et al. showed that polyQ-htt activates the JNK pathway and induces JNK3 activity in HD models, leading to fast axonal transport inhibition through phosphorylation of kinesin-1 motor protein, which induces microtubule dissociation [70]. Thus, the JNK pathway participates in HD pathology. Kinase-inhibitor treatments in a rat model of HD reduced the extension of the brain-lesion area by blocking the JNK pathway and conferred neuroprotection [71]. This approach provides a new target to prevent progressive HD neurodegeneration; however, further studies are required to better understand the contribution of the JNK pathway to HD progression.

In conclusion, JNK inhibitors have been proposed as new potential candidates for neurodegenerative diseases, but their lack of specificity and unknown toxicity profile have limited the positive effect of these molecules in clinical trials. However, recent research has focused on the JNK isoform structures, and the development of new inhibitors with selective affinity for specific isoforms will facilitate the clinical target of the JNK pathway in multiple diseases.

## 4. JNK in Regeneration/Repair after an Injury to the CNS

CNS damage produced by brain stroke, spinal-cord injury or neurodegenerative diseases often results in permanent disabilities due to the limited regenerative capacity of this tissue. Several studies have supported the importance of JNK signalling in response to nerve injuries in both degeneration and repair. In the injured axons, retrograde transport of JNK can alter somal transcription of the injury-response molecules ATF3 and Hsp27, important for axonal outgrowth [72,73]. On the other hand, genetic or pharmacological inhibition of JNK signalling in multiple models of axonal injury delayed axonal degeneration [74,75], indicating that JNK is also required for axonal degeneration. Moreover, upon nerve injury of a rat model, JNK3 inhibited axonal growth through the interaction with the Kluppel-like transcription factor 9 (KLF9) [76]. JNK signalling also activates members of the Bcl-2 family, triggering apoptosis after injury or stress [77,78,79].

MAP3K is one of the first members of the JNK pathway. It is a dual-leucine zipper kinase (DLK), a conserved kinase with orthologues in mammals (MAP3K DLK), *Drosophila* (DLK/Wallenda) and *C. elegans* (DLK-1) [80]. MAP3K is an important axon-injury sensor [81]. DLK–JNK signalling contributes in multiple ways to axonal injury response and axonal regeneration. The DLK/Wallenda protein is present in axons, and protein levels are increased in response to axonal injury in *Drosophila* [82,83]. JNK-dependent phosphorylation of DLK is required for the stabilization of DLK levels [84]. DLK regulates microtubule stability [81,85,86], essential for axonal regeneration in spinal-cord injury [87]. DLK/Wallenda overexpression has also been shown to be protective in *Drosophila* motor neuron axons [83]. DLK is also an essential molecule for the injury-dependent activation of the retrograde transport of p-STAT3 to the cell body, necessary for the activation of the neuronal regenerative program in mice [88]. Moreover, it has been demonstrated that in *C. elegans*, DLK-1 is both necessary and sufficient for injury-induced autophagy activation; DLK-1 limits the levels of LIN-12 and NOTCH proteins, suggested to promote axon regeneration [89].

### 4.1. JNK Signalling in Glial Cells upon Injury

All the evidence summarized above demonstrates the importance of JNK signalling in injured axons; however, JNK-signalling activation also occurs in glial cells in response to injury. Different injury paradigms in *Drosophila* showed a consistent activation of AP-1 transcription in glial cells upon injury, downstream of JNK. Unknown molecules from axonal debris activate the engulfment receptor Draper, which in turn regulates *draper* transcriptional upregulation via STAT92E [90] and the JNK pathway [74]. Draper is a conserved receptor with orthologues in mammals (MEGF10) and *C. elegans* (CED-1), and it is required for engulfment in several cell types, including germ line, epithelial cells, microglia and astrocytes [91,92,93,94].

In *Drosophila* and *C. elegans,* downregulation of JNK reduces axon regeneration and provokes axon debris accumulation following injury. In the injury context, JNK also activates MMP-1 expression [95], which is required for glial cells to infiltrate in the injured tissue and remove cell debris via Draper. Glial-specific overexpression of *draper* is sufficient to rescue engulfment defects associated with the loss of JNK signalling [74]. In addition, knockdown of the JNK pathway components blocks CED-1 mediated axon regeneration and axon debris removal [96], which suggests a role for JNK in Drpr/CED-1-mediated axon regrowth.

### 4.2. JNK in Neurogenesis and Regeneration

Finally, JNK could also trigger CNS regeneration by promoting neuronal differentiation from neural stem cells (NSCs). So far, neurogenesis in mammals is restricted to two major regions, the subependymal zone (SEZ) and the subgranular zone (SGZ) of the dentate gyrus, where adult NSCs are present, but the neuronal differentiation rate of NSCs is limited [97]. Therefore, a promising strategy to boost CNS regeneration is to find candidates to promote neural differentiation from NCSs. Noncanonical Wnt signalling such as the Wnt/JNK pathway has a positive effect on neuronal differentiation in cell culture and during development [98,99,100]. Moreover, it has been recently shown that Wnt5a upregulates *miRNA200b-3p* expression through MAPK/JNK signalling, and *miRNA200b-3p* suppresses the RhoA/Rock signalling required for neuronal differentiation. Thus, Wnt5a promotes NSC differentiation into neurons, and more remarkably, transplantation of NSCs overexpressing Wnt5a results in tissue repair and locomotor functional recovery in rats after spinal-cord injury [97]. However, other studies have revealed that different JNK isoforms are involved in the adult neurogenesis control in a different manner. For instance, inhibition of JNK1 or mice lacking JNK1 show an increased number of neural progenitors in the SGZ of the hippocampus, whereas the absence of JNK3 reduces it [101,102]. These results again point out the importance of finding drugs with high specificity for the different JNK isoforms to treat CNS disorders.

## 5. Conclusions

The JNK pathway is involved in multiple biological features and plays a key role in the CNS. The complexity of this signalling pathway is revealed by the opposite functions of JNK signalling depending on the tissue and the cellular environment. Hence, it is difficult to reach a clear-cut understanding of the contributions of this kinase cascade to the cell physiology. However, here we reviewed some of the most important roles of the JNK pathway in CNS pathologies, and as a potential pharmacological target to prevent its consequences. In this context, CNS cells trigger a stress response that increases JNK levels, which leads to a predominant cell-death program in neurodegenerative events, tumour-aggressive properties in glioblastoma progression and neurodegeneration or promotion of axonal regeneration upon an injury. All the evidence described here, with the conservation of JNK-pathway signalling in organism models such as *Drosophila melanogaster* or *C. elegans,* will help to decipher the multiple JNK activities and open promising clinical perspectives to treat CNS pathologies.

## Figures and Tables

**Figure 1 ijms-22-03883-f001:**
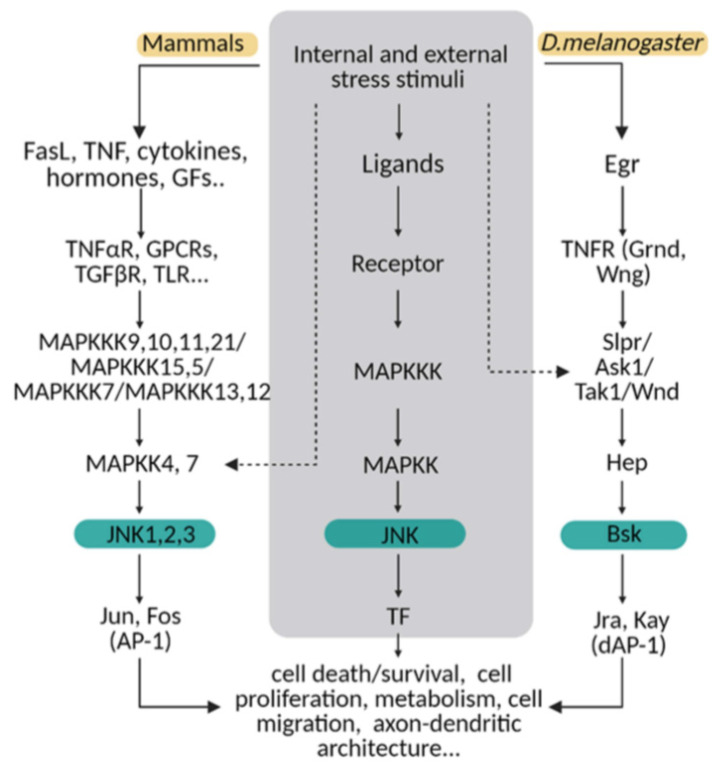
The JNK pathway is conserved between mammals and flies. Upon internal and external stress stimuli, ligands (ex. TNF/Egr) activate transmembrane receptors (ex. TNFαR/ Grnd) and initiate a cellular response based on a MAPK cascade phosphorylation. In mammals, different ligands can trigger the signal path; while in *Drosophila,* Eiger (Egr) is the TNF ligand that predominantly initiates the stress response via TNF receptors (Grindewal, Grnd; or Wengen, Wng). In mammals, there is MAPK path gene redundancy that is shown in this diagram compared with flies. MAPKs are encoded by multiple genes in mammals, whereas in *Drosophila,* one single gene is described for each enzyme. The cellular response converges in JNK/Bsk phosphorylation, which triggers the expression of transcription factors as Jun/Jra and Fos/Jay, known as an AP-1/dAP-1 complex. Finally, AP1/dAP-1 modulates the transcriptional program of genes involved in a variety of biological activities. This pathway can be activated at other steps indicated with dotted lines in the diagram.

**Figure 2 ijms-22-03883-f002:**
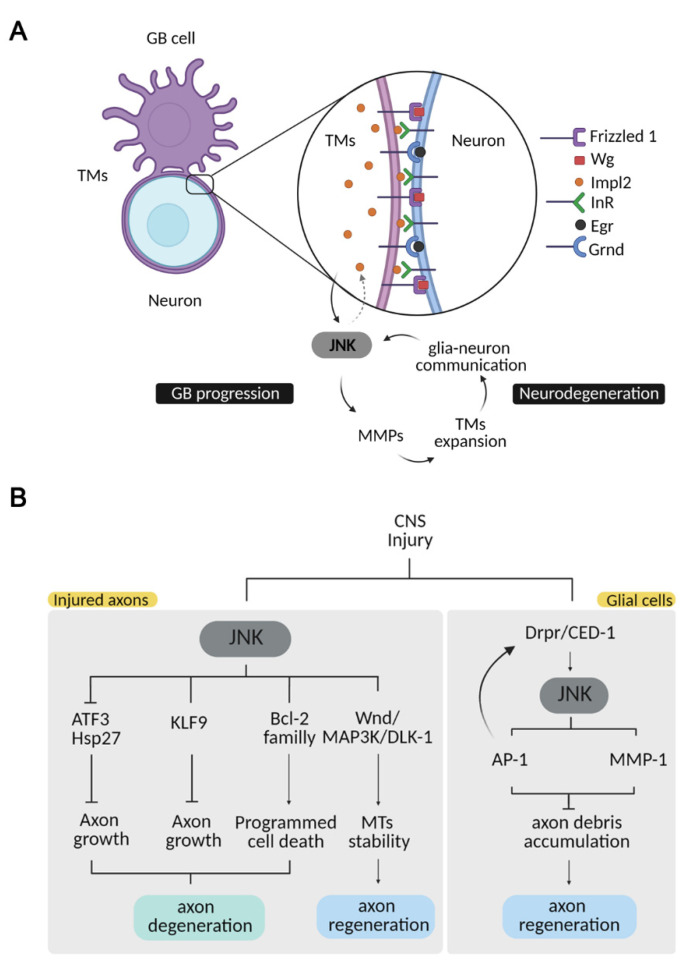
JNK pathway in CNS pathologies. (**A**) JNK integrates a positive loop promoting tumour growth and neurodegeneration. GB cells extend a TM network wrapping the surrounding neurons, promoting glia–neuron signalling communication. TMs filaments deplete neuronal Wg/WNT and Egr/TNF and relocalize Frizzled 1 and Grnd to activate the JNK pathway in tumor cells. JNK activation leads to MMP transcription, required for tumor infiltration into the healthy tissue, which expands the TM network, thereby describing a positive feedback loop. As a result, neurons degenerate. Neurodegeneration is also reinforced by the JNK-Impl2-InR signal in neurons, which alters mitochondrial metabolism and facilitates synapse loss and neuronal death. (**B**) Neuronal and glial JNK mediates different pathways upon CNS injury. JNK activation in injured axons promotes axon degeneration through the activation of downstream molecules such as ATF3/Hsp27 and KLF9, which in turn inhibit axonal growth or members of the Bcl-2 family to trigger apoptosis. JNK activation in injured axons also promotes microtubule stability and axonal growth through Wnd/MAP3K/DLK-1 phosphorylation. JNK activation in glial cells upon injury is triggered by Draper, and it is required for further Draper activation, as well as for cell-debris removal. JNK activates MMP-1, necessary for glial infiltration for cell-debris clearance. This phagocytic response is necessary for axonal regeneration.

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
