# Peer review of "JNK Pathway in CNS Pathologies"

_ijms, 2021, doi:10.3390/ijms22083883_

Round 1

Reviewer 1 Report

Overall the manuscript is a good review of recent knowledge advances regarding the (sometime conflicting) roles of JNK in disease and healing within the nervous system. In general the information is well presented, and is appropriate for publication in this journal.

I have a few comments. First, I think it should be a bit more clearly delineated when the authors switch back and forth between the fly and mammalian data. This sometimes became confusing, particularly in the introduction.   

Figure 2 seems to have not uploaded or integrated into the manuscript correctly, leaving me unable to evaluate it. I can only see a fraction of the figure.

Finally, the manuscript requires extensive editing for standard English grammar, spelling, and to correct incorrect usage of set phrases (e.g. two faced rather than two fated). 

Author Response

Overall the manuscript is a good review of recent knowledge advances regarding the (sometime conflicting) roles of JNK in disease and healing within the nervous system. In general the information is well presented, and is appropriate for publication in this journal.

We would like to thank the reviewer for his/her positive comments

I have a few comments. First, I think it should be a bit more clearly delineated when the authors switch back and forth between the fly and mammalian data. This sometimes became confusing, particularly in the introduction.

We have included further details on the specific animal model used in each case, and the conserved mechanisms through evolution  

Figure 2 seems to have not uploaded or integrated into the manuscript correctly, leaving me unable to evaluate it. I can only see a fraction of the figure.

We have uploaded the figure again

Finally, the manuscript requires extensive editing for standard English grammar, spelling, and to correct incorrect usage of set phrases (e.g. two faced rather than two fated).

We have reviewed the grammar, and the text has been reviewed by a native english scientist.

Reviewer 2 Report

The manuscript ijms-1137233, JNK pathway in CNS pathologies, is presenting an interesting research theme, but fails to deliver what is expected of a review paper: an expert's assessment of the recent major discoveries and the knowledge status of this particular field.

Overall the style of the paper is hard to understand. The authors add various data on JNK pathway without a clear and logical continuity. There is a need for a clear objective of all information presented. The paper lacks of any discussion for potential new research, ideas of where this field will develop.

The references are in their majority outdated and their presentation is not according to the journal’s style. The main objective of any review paper should be to present the most recent major advances and discoveries in that field. There are large sections based on only 2 or 3 references.

row 4, the authors’ affiliation should be done according to the journal’s editorial style

In the section 25-35 the authors present the CNS homeostasis mechanisms. Is it the UV irradiation a mechanism of JNK activation in the CNS?

Row 34-35. There are only old references. Please update some of the references.

Row 38, check the editing

Even if the authors mention some aspects on Drosophila on the “JNK in glioblastoma tumor progression” section, in my opinion the section 36-72 is not relevant and can confuse the readers. Why is important in the description and analysis of the JNK pathways in CNS pathologies to describe the case of Drosophila? This section should be removed and I advise the authors to focus on the subject of CNS pathologies. The same argument for Figure 1. Present only the activation pathway relevant to CNS pathologies.   

The section 86-100 needs to be trimmed of all the irrelevant data. There is no real need to compare “JNK pathway between mammals and insects”

The section 2. JNK in glioblastoma tumor progression presents only a collection of data on JNK without any clear objective. The authors should try to offers some answers to the questions like what are the causes behind glioblastomas? Are there preventive measures that can be taken in correlation with JNK activation? Are the drug therapies based on JNK inhibition targeting glioblastomas? Some future perspectives?

Figure 2 is truncated and therefore difficult to interpret its correctness.

Row 220, it is a very simple statement with no input from the authors. Are there inhibitors of JNK in clinical studies as treatment for AD? Do the authors consider that the inhibition of JNK could be a solution for future AD therapies? Could the inhibition of JNK target secondary undesirable effects?

The subsections on Parkinson and Huntington disease are seriously underdeveloped.

The paper has a great potential to become important for this field, but there is a lot of work needed for the authors to improve it and to add it real value.

Author Response

The manuscript ijms-1137233, JNK pathway in CNS pathologies, is presenting an interesting research theme, but fails to deliver what is expected of a review paper: an expert's assessment of the recent major discoveries and the knowledge status of this particular field.

Overall the style of the paper is hard to understand. The authors add various data on JNK pathway without a clear and logical continuity. There is a need for a clear objective of all information presented. The paper lacks of any discussion for potential new research, ideas of where this field will develop.

We have reorganized the text completely, and added new information and references

The references are in their majority outdated and their presentation is not according to the journal’s style. The main objective of any review paper should be to present the most recent major advances and discoveries in that field. There are large sections based on only 2 or 3 references.

We have included further references in all the sections, however, there are some sections as the contribution of JNK to HD that do not have any recent publication in the literature.

row 4, the authors’ affiliation should be done according to the journal’s editorial style

We have changed this accordingly

In the section 25-35 the authors present the CNS homeostasis mechanisms. Is it the UV irradiation a mechanism of JNK activation in the CNS?

JNK is activated in other tissues by UV light, however there are no specific reports in the CNS. We have modified the sentence accordingly to avoid misunderstandings.

Row 34-35. There are only old references. Please update some of the references.

We have included new recent references

Row 38, check the editing

We have checked it

Even if the authors mention some aspects on Drosophila on the “JNK in glioblastoma tumor progression” section, in my opinion the section 36-72 is not relevant and can confuse the readers. Why is important in the description and analysis of the JNK pathways in CNS pathologies to describe the case of Drosophila? This section should be removed and I advise the authors to focus on the subject of CNS pathologies. The same argument for Figure 1. Present only the activation pathway relevant to CNS pathologies.   

We have reduced this section but we have not removed it completely. The relevance of Drosophila in particular is noticeable, and in particular, JNK pathway contribution to glioblastoma progression has been recently documented and brings novel perspectives to the field. Therefore, we would like to maintain this section.

Figure 1 aims to show the key players in JNK pathway, and the conservation between Drosophila and mammals, two of the most used animal models to determine the mechanisms behind JNK pathway function. In general, we show the factors that have a relevance for JNK pathway under pathological conditions.

The section 86-100 needs to be trimmed of all the irrelevant data. There is no real need to compare “JNK pathway between mammals and insects”

We have trimmed this section accordingly. We believe that the study of JNK pathway has historically been supported by mammals and Drosophila. We do not understand why the information in this section seems irrelevant if these are the main sources of knowledge for this pathway.

The section 2. JNK in glioblastoma tumor progression presents only a collection of data on JNK without any clear objective. The authors should try to offers some answers to the questions like what are the causes behind glioblastomas? Are there preventive measures that can be taken in correlation with JNK activation? Are the drug therapies based on JNK inhibition targeting glioblastomas? Some future perspectives?

We have included new relevant information regarding the potential of JNK as a therapeutic target in GB

Figure 2 is truncated and therefore difficult to interpret its correctness.

We have uploaded it again

Row 220, it is a very simple statement with no input from the authors. Are there inhibitors of JNK in clinical studies as treatment for AD? Do the authors consider that the inhibition of JNK could be a solution for future AD therapies? Could the inhibition of JNK target secondary undesirable effects?

Yes, there are JNK inhibitors to target AD (Hepp Rehfeldt et al., 2020; Yarza et al., 2016). We have included discussion about the potential contribution of JNK inhibitors as a target

The subsections on Parkinson and Huntington disease are seriously underdeveloped.

We have expanded both sections accordingly, however, there is a lack of recent information about HD and JNK.

The paper has a great potential to become important for this field, but there is a lot of work needed for the authors to improve it and to add it real value.

We have done a massive change to the manuscript following your suggestions, we hope that this new version fulfills your expectations

Round 2

Reviewer 2 Report

The authors changed significantly their manuscript and improved its value. It can be published after some small editorial corrections.

Author Response

We have included all the following suggestions:

Point 1.- In line 171 and 204 the authors talk about glioblastoma samples and more specifically (EGFR-glioblatoma sample). It would be interesting to define this concept. For example, how are obtained and processed.

We have modified this paragraph accordingly now in lines 83-93 and included further references Li et al. 2008

Point 2. Figure 2 is still incomplete.

We have uploaded a new version of Fig 2

Point 3.  In the 3 section" JNK in neurodegenerative disorders", I suggest to make subtitles for every neurodegenerative disease.

We have included subtitles

Point 4. The explanation about JNK and HD, would be reorganized. I suggest to start explaining that in HD there are alterations in neuronal transport and then explain how JNK pathway controls this cellular process.

We have updated this section accordingly

Point 5. In the section 4 "JNK in regeneration repair after an in jury in the CNS", I suggest making a subtitle: at the beginning of the paragraph, saying: JNK signalling has a role in axonal injury response and axonal regeneration and then justifies this sentence.

 Line 220 now states that: Several evidences support the importance of JNK signaling in response to nerve injuries in both degeneration and repair

Another subtitle:  the relation of JNK signalling in glial cells after injury and finally a subtitle before explaining JNK, regeneration and stem cells.  (Mol Neurobiol 2019 Aug;56(8):5856-5865 is an article that evidence the role of JNK isoforms in the neurogenic activity of cells located in the SGZ of the hippocampus).

We have included these sections and this reference (line 281)

We have extensively revised and corrected the English language and style following the suggestions of the reviewer. A native professional scientist has contributed to these corrections.